# Synthesis and Characterization of SiO_2_-Based Graphene Nanoballs Using Copper-Vapor-Assisted APCVD for Thermoelectric Application

**DOI:** 10.3390/nano14070618

**Published:** 2024-04-01

**Authors:** Nurkhaizan Zulkepli, Jumril Yunas, Muhammad Aniq Shazni Mohammad Haniff, Mohamad Shukri Sirat, Muhammad Hilmi Johari, Nur Nasyifa Mohd Maidin, Aini Ayunni Mohd Raub, Azrul Azlan Hamzah

**Affiliations:** 1Institute of Microengineering and Nanoelectronic (IMEN), Universiti Kebangsaan Malaysia (UKM), Bangi 46300, Selangor, Malaysia; khaizan2821@uitm.edu.my (N.Z.); aniqshazni@ukm.edu.my (M.A.S.M.H.); mshukrisirat@gmail.com (M.S.S.); p111586@siswa.ukm.edu.my (M.H.J.); p103661@siswa.ukm.edu.my (N.N.M.M.); p109011@siswa.ukm.edu.my (A.A.M.R.); 2Centre of Foundation Studies, Universiti Teknologi MARA, Cawangan Selangor, Kampus Dengkil, Dengkil 43800, Selangor, Malaysia; 3Research Center for Advanced Materials, National Research and Innovation Agency (BRIN), South Tangerang 15314, Banten, Indonesia; dedi011@brin.go.id

**Keywords:** graphene nano balls, copper-vapor-assisted, atmospheric pressure chemical vapor deposition, thermoelectric, bismuth telluride

## Abstract

This study describes a method by which to synthesize SiO_2_-based graphene nanoballs (SGB) using atmospheric pressure chemical vapor deposition (APCVD) with copper vapor assistance. This method should solve the contamination, damage, and high costs associated with silica-based indirect graphene synthesis. The SGB was synthesized using APCVD, which was optimized using the Taguchi method. Multiple synthesis factors were optimized and investigated to find the ideal synthesis condition to grow SGB for thermoelectric (TE) applications. Raman spectra and FESEM-EDX reveal that the graphene formed on the silicon nanoparticles (SNP) is free from copper. The prepared SGB has excellent electrical conductivity (75.0 S/cm), which shows better results than the previous report. Furthermore, the SGB nanofillers in bismuth telluride (Bi_2_Te_3_) nanocomposites as TE materials exhibit a significant increment in Seebeck coefficients (S) compared to the pure Bi_2_Te_3_ sample from 109 to 170 μV/K at 400 K, as well as electrical resistivity decrement. This approach would offer a simple strategy to improve the TE performance of commercially available TE materials, which is critical for large-scale industrial applications.

## 1. Introduction

It is becoming more critical to comprehend the synthesis and characteristics of low-dimensional carbon nanomaterials such as graphene, carbon nanotubes (CNTs), and fullerenes on semiconductor materials owing to their synergetic effects [1,2,3]. The combination of these heterogeneous materials has broad potential applications, from environmental studies and construction materials to portable electronic devices, highlighting the tremendous potential implementation of graphene-based nanocomposite for future power generation [4] On top of that, graphene, CNTs, fullerenes, and their derivatives have garnered worldwide attention in the past decade, offering enormous potential in biosensors [5,6,7], flexible pressure sensors [8], nanocomposite electrodes [9,10], tunable absorbers [11,12], and photocatalysts [13,14], as well as energy generation [15]. 

Interest in the direct growth of graphene on semiconductors has been intriguing in recent years due to the impressive improvements of graphene–semiconductor interfaces, which have contributed to synergetic effects in hybrid devices. As a result, researchers began investigating more innovative approaches for directly growing graphene on silicon surfaces, including via hydrothermal carbonization [16], pulsed laser ablation [17], and dispersion polymerization technique [18]. On the other hand, many research groups continue their work on graphene–silicon nanocomposites synthesis for lithium-ion batteries using spray pyrolysis [19] and high-energy ball milling [20].

Chen et al. [21] reported their study on growing high-quality polycrystalline graphene on quartz by chemical vapor deposition (CVD). By using this technique, no post-growth transfer procedure is required, while remaining compatible with existing silicon manufacturing processes. Despite the extensive research efforts to directly grow graphene on silicon surfaces, the primary obstacles remain the mass production of high-quality graphene with few or no impurities or defects and the ability to produce graphene with a large grain size at a reasonable cost [22].

After all, CVD is one of the most cost-effective and scalable techniques for producing graphene with high crystallinity [22]. In the CVD process, there are two types of total reaction pressure: low-pressure CVD (LPCVD) [23] and atmospheric-pressure CVD (APCVD). Pumping systems are eliminated in APCVD, making it an attractive solution for low-cost graphene synthesis [24]. CVD methods have been extensively investigated in the literature to synthesize graphene on SiO_2_, as shown in Table 1.

The use of metals as catalysts to effectively accelerate the decomposition of carbon sources, such as copper or nickel, is common. Vaporous copper atoms were evaporated from copper foil owing to the temperature commonly used in the annealing process (1000 °C) being slightly lower than the melting point of copper (1085 °C) [25]. The floating copper atoms operate as catalysts, allowing processes that would otherwise be prevented or hindered by a kinetic barrier in direct catalysis on oxide surfaces to proceed. The distant metal-catalyzed decomposition of hydrocarbons resulted in the formation of graphene with minimal defects and amorphous carbons. Providing optimal copper particles may result in more successful reactant collisions, boosting the efficiency of carbon feedstock decomposition in the gas phase.

As shown in Table 1, studies on graphene growth on SNP by the CVD method are lacking in the main literature. CVD methods were successfully exploited by Son et al. [26] to grow graphene on SNP. Graphene balls, consisting of 3D graphene layers surrounded by a SiOx nanoparticle, were reported, which are composed of graphene and silicon nanoparticles at its centre. They demonstrated the CVD process by utilizing a fixed-bed vertical tube reactor, demanding high precision and complexity. Recently, Zhao et al. [27] displayed graphene ball potential to improve the properties of stress sensors by dispersing it with graphene oxide through a one-step hydrothermal reduction self-assembly method.

Graphene synthesis by CVD is influenced by a wide range of variables, including temperature [28], pressure [29], and growth and annealing time [30]. As a result, setting up the ideal conditions with numerous factors is very time-consuming. Taguchi method offers a practical and systematic strategy for optimizing the system in such conditions [31]. Using orthogonal arrays, it is possible to analyze many variables with a short number of trials using an empirical method that combines mathematical and statistical approaches [32]. Primary parameters may thus be thoroughly studied, while secondary ones can be pushed aside. As a result, valuable time, energy, and resources may be spared. This method has been previously applied to optimize the production of graphene from various carbon sources such as methane, ethanol, plastic waste [33], and oil palm fibre [34].

Graphene has been widely utilized in electronic applications due to its high electrical conductivity (10^6^ S/cm). TE materials should exhibit both high electrical conductivity and extremely low thermal conductivity values in order to qualify as good TE materials [35]. Integrating graphene with SNP preserves unique traits of the two-dimensional system, including a large specific surface area, elasticity, and high conductivity, while retaining excellent characteristics of SNP that are stable chemically and easy to modify, high-temperature resistance, and low cytotoxicity [36]. Applying uniform graphene coating to the SNP using a CVD method can potentially increase electrical conductivity while preserving low thermal conductivity values.

In this work, the SGB was directly grown on SiO_2_ nanopowder (SNP) and optimized using the Taguchi method for TE applications. The SGB was grown using the APCVD process with copper vapor as the catalyst. The key characteristic of the proposed method is hydrocarbon breakdown through floating metal catalysis. Then, the optimized SGB were uniformly mixed in TE material nanocomposites, and its effect on the TE properties was evaluated.

**Table 1 nanomaterials-14-00618-t001:** Growth of graphene on SiO_2_ with the CVD method.

SiO_2_ Surface	Gas	Flow	Pressure	Temperature	Growth Time	Catalyst	Ref.
		(sccm)	(Torr)	(°C)	(min)		
Flat substrate	Ar/H_2_/CH_4_	65:50:14	Atmospheric	1100	180	O_2_	[21] 2011
Flat substrate	Ar/H_2_/CH_4_	230:5:30	Atmospheric	1000	30	Cu	[37] 2012
Flat substrate 300 nm	H_2_/CH_4_	20:30	5	1000	30	Cu	[38] 2013
Flat substrate Quartz	Ar/H_2_/CH_4_	100:20:50	60–360	950–1100	5	Ni	[25] 2014
Nanoparticles 20–30 nm	CH_4_	50	Atmospheric	1000	60	-	[26] 2017
Flat substrate Si 500 μm	H_2_/CH_4_	100:5	0.007	1100	60	Cu	[39] 2018
Nanoparticles 20–30 nm	Ar/CH_4_	200:25	Atmospheric	1000	20	-	[27] 2019
Flat substrate 300 nm	Ar/CH_3_OH		0.15	1020	30	Cu	[40] 2019
Flat substrate 300 nm	H_2_/CH_4_	25:15	0.00007	950	60	-	[41] 2020
Nanoparticles 20–30 nm	Ar/H_2_/CH_4_	78:2:50	Atmospheric	1000	60	Cu	This work

## 2. Experimental Section

### 2.1. Graphene Growth

SGB was directly grown by using the APCVD method. Copper foil, C_S_ (99.9% purity; 50 µm thick) varied in size, is initially sonicated in ethanol for 5 min, distilled water and IPA for 5 min, and lastly blown dry with nitrogen. The sample is immediately deposited on a porcelain boat and placed into a horizontal quartz tube with an inner diameter of 25.4 mm and a length of 800 mm at room temperature. Over 0.1 g of quartz wool (QW) and SNP (average diameter: 20–30 nm, Alfa Aesar, Haverhill, MA, USA) were placed into the same porcelain boat. The amount of SNP that was loaded into each batch was 0.01 g. The distance between copper foil and SNP, C_D_, was varied as shown in Figure 1a.

The quartz tube was sealed and purged with 80 sccm Ar/H_2_ for 30 min at atmospheric pressure before graphene growth could begin. This step was to flush the air from the quartz tube. While Ar/H_2_ gas flowed through the system, it was heated to 1000 °C at a ramping rate of 33 °C/min. The copper foil was annealed at 1000 °C for 30 min under Ar/H_2_ atmosphere to improve the copper grain size and to assure the elimination of native oxide and a smooth copper surface. Graphene was then grown by exposing it to an Ar/H_2_ and CH_4_ mixture at varying flow rates, Φ_CH4_, for 60 min. After growth, CH_4_ was shut off for the cooling step. As demonstrated in the growth protocol in Figure 1b, the samples were cooled from 1000 °C to room temperature in an Ar/H_2_ ambient.

A design of experiment (DOE) was conducted using Minitab 17 and the Taguchi Method to find the best synthesis parameter. This study evaluated the most relevant factors in the copper-catalysed synthesis of graphene. Three major parameters were reported to be the most influential: (1) methane flow rate (Φ_CH4_); (2) copper foil size (C_S_); and (3) copper distance (C_D_) [42]. The considered factors and their levels are shown in Table 2. Orthogonal arrays reduce the number of experiments by allowing randomized runs. The analysis aimed to obtain a low I_D_/I_G_ ratio of SGB.

### 2.2. Characterization

The SGB was characterized using room temperature confocal micro-Raman imaging spectroscopy (DXR2Xi, Thermo Scientific, Waltham, MA, USA) to confirm the existence of graphene on SNP. The excitation wavelength was 532 nm, while the exposure time was 0.1 s. These measurements were conducted directly on the SGB. Three discrete points were investigated to characterize each sample and exclude potential spatial inhomogeneities correctly. The spectra were then averaged, normalized, and analyzed using spectroscopic analysis software. The morphological and cross-sectional views of the SGB were observed by field emission scanning electron microscopy (FESEM, Zeiss Merlin, Jena, Germany) and transmission electron microscopy (TEM, Thermo Scientific, Talos L120C, 120 kV). X-ray photoelectron spectroscopy (XPS) was carried out using an ULVAC-PHI Quantera II instrument with an Al-Kα monochromatic source (hv = 1486.6 eV) operating at 15 kV and 50 W. A Hall-effect measuring instrument (Ecopia, HMS-5300, Anyang, Republic of Korea) was utilized to characterize the electrical conductivity and charge mobility of the SGB for the electrical experiments. Electrical characterization was performed on a 0.5 cm × 0.5 cm sample size using a typical van der Pauw method under ambient air at 300 K by a Hall effect measurement system. The SGB was suspended in ethanol and was drop-casted on the surface of a glass substrate. The electrical resistivity, ρ, and Seebeck coefficient, S, of the SGB and TE material nanocomposites were tested using the LSR-4/800 (Linseis, Bayern, Germany) under low pressure (10^−2^ torr). Rectangular bars cut from 12 mm pellets with dimensions of 11.5 mm × 2.4 mm × 2.4 mm were tested at numerous temperature points ranging from room temperature to 500 K.

## 3. Results and Discussion

### 3.1. Optimization of SGB Synthesis Process

The important characteristic of the synthesized graphene layer is the I_D_/I_G_ ratio, which is determined through Raman spectroscopy analysis on the sample. Initially, nine samples were synthesized using the copper-vapor-assisted APCVD method, and the extracted I_D_/I_G_ ratio was calculated. As shown in Figure 2a, a small I_D_/I_G_ ratio via Raman spectroscopy determines the good crystallinity of carbon-based materials as it indicates a low defect density [43]. 

To find the optimum process parameter, the I_D_/I_G_ ratio by the nine synthesized samples was analyzed using Taguchi analysis. The experimental data were converted into a signal-to-noise ratio (SNR) to study each factor’s effect on SGB crystalline quality. In this work, the SNR was determined based on the “smaller is better” model in order to decrease the defect density in the graphitic lattice. The factor with the highest SNR value influences the considered output the most [32,44].

Figure 2b shows the mean of SNR ratios for each factor. According to the Taguchi analysis, the C_D_ had the most significant impact on defectiveness in SGB, while Φ_CH4_ was the second most influential. The SGB synthesized under the conditions at Φ_CH4_ = 50 sccm, C_S_ = 2 × 3 cm, and C_D_ = 5 cm had the highest SNR, specifying the optimum process conditions for SGB synthesis. All these optimum values are indicated by the red circles. This study determines the best SGB synthesis factors for the lowest defect density.

### 3.2. Characterization of the Optimized SGB

#### 3.2.1. Raman Analysis

The Raman spectra of SGB grown using the copper-vapor-assisted APCVD method are shown in Figure 3a. A combination of three prominent peaks can be seen in the Raman spectra of graphene: (1) The D-band, around 1350 cm^−1^, gives an excellent indication for measuring the level of disorder present in the sample; (2) the G-band, around 1580 cm^−1^, is related to the in-plane vibration of sp^2^-hybridized carbon atoms; and (3) the 2D band, roughly situated at 2680 cm^−1^, represents the thickness of graphene [45]. The analysis of the I_D_/I_G_ ratio is a well-known method for evaluating the defect density of a graphene sample, whereas the I_2D_/I_G_ ratio provides information on the number of layers present in a graphene sample. The method enables the differentiation of graphene from graphite and the assessment of the quality and thickness of graphene [46]. Defects in graphene may impair charge carrier mobility and hence electrical conductivity, which is detrimental to electronics applications [47].

In this work, the 2D band of SGB centred at 2684 cm^−1^. The I_D_/I_G_ ratio indicates that the deposited SGB were mostly made of defective graphene layers. Defects are frequently associated with Stone–Wales defects, the presence of five or seven rings that alter the regular six-ring structure in a graphene lattice [47]. The I_2D_/I_G_ ratio of the SGB indicates the development of graphene multilayers.

A series of experimental studies were conducted to determine the effect that copper played in the formation of graphene on SNP. For example, removing copper or replacing copper with Ni in the growth results in either amorphous carbon or a higher defect density of graphene on the SNP. As shown in Figure 3b, Ni is known to have a much higher melting point (mp = 1455 °C) than copper (mp = 1084 °C); hence, replacing copper ith Ni results in a higher defect density of graphene, as indicated by higher I_D_/I_G_ ratios. Figure 3c shows that the exact process without copper foil produces no graphene. Compared to the process with the Cu catalyst, there are no clear 2D peaks observed. This observation proves that the distantly given copper vapor does indeed cause graphene growth, as shown in Figure 3c.

A high-temperature system is required for the growth process since evaporated copper is required to form graphene on SNP. Due to its high vapor pressure, copper foil rapidly evaporates to give off copper vapor at a growth temperature of 1000 °C [37]. Subsequently, copper vapor interacts with methane precursors adsorbed on the SNP, resulting in the nucleation of graphene formation in the lateral direction. The optimal ratio of copper vapor to methane gas (Cu_vapor_/CH_4_) seems to have contributed a significant role in the effective synthesis of SGB.

#### 3.2.2. Morphology Analysis

Another essential requirement for the copper vapor-assisted CVD process is that the final SGB be copper-free. For this purpose, energy-dispersive X-ray (EDX) spectroscopy was employed to detect the amount of copper residue present in the graphene grown on SNP. To further ascertain that graphene growth is uniformly distributed on SNP, FESEM-EDX elemental mapping analysis of SGB has been carried out. Figure 4a shows FESEM images of SGB and the elemental maps of silicon, oxygen, and carbon, along with the corresponding overlay shown in Figure 4b. The SGB was composed of three elements: 45.1% of C; 23.6% of O; and 31.3% of Si, where C refers to graphene and O and Si correspond to SNP. These images reveal that graphene is uniformly distributed on SNP. Neither the EDX nor the FESEM-EDX elemental mapping analysis of SGB showed the presence of copper residues within the detection limit. This finding suggests that no copper-related chemical species are detected in the samples, suggesting that the copper acts as a catalyst in the synthesis process but is not integrated into the SGB [48].

The morphologies of the SGB are shown in Figure 5. Figure 5a represents SNP before graphene growth, while Figure 5b illustrates the SGB after the graphene growth process. The TEM micrographs of SNP in Figure 5a show that they were spherical-shaped particles, about 20–30 nm in size, and prone to aggregation. Graphene as a transparent sheet can be seen in Figure 5b, and the thickness of these layers is approximately 20–40 nm. The figure shows distinguished multilayers of graphene and SNP. An interesting point to note here is that the 3D sheet morphology is visible in Figure 5c. The graphene that enwraps the spherical SNP surface is revealed as being transparent multilayers, which is further confirmed by the selected area of electron diffraction (SAED) pattern, as shown in Figure 5d.

#### 3.2.3. Chemical Composition

XPS measurements were performed to determine the chemical composition of the synthesized GBS at optimum conditions. Figure 6a shows the survey spectrum of SGB in which the presence of elements C 1s, Si 2p 2s, and O 1s can be observed. Figure 6b shows high-resolution XPS spectra for C sp^2^, sp^3^. Figure 6b shows the XPS analysis of graphene grown on the surface of SNPs. The presence of graphene was confirmed at the peak of 284.5 eV in the C–C characteristic group, indicating sp^2^ carbon bonding [26]. This XPS analysis aims to confirm that the SGB has graphene atomic bonds. Since graphene has high electrical conductivity, it is important to confirm that the SGB has an sp^2^ C–C bond.

#### 3.2.4. Electrical Characterization of SGB

The Hall mobility (μ_Hall_) of the SGB reaches up to 550 cm^2^/Vs, with an average Hall effect coefficient (R_Hall_) of 4.89 cm^3^/C and electrical conductivity of 75 S/cm. The positive value of the average R_Hall_ indicates that holes are the predominant charge carriers in the SGB. The R_Hall_ value is comparable to graphene grown on metal catalysts [25,40]. For comparison with the previous CVD method, our SGB exhibits a much higher average electrical conductivity compared to those of SGB grown via CH_4_ as the only gas during the growth process (∼40 S cm^−1^) [26] and graphene nanoballs grown with 20 min growth time (∼1 S cm^−1^) [27].

### 3.3. Material Evaluation for TE Application

In this work, the performance of SGB as nanofillers to improve the TE properties of TE generators was studied. TE generators’ efficiency is determined by the material’s performance, which is directly connected to the figure of merit, $\mathrm{ZT}=S^{2}T/\rho \kappa$, where S, ρ, κ, and T denote the Seebeck coefficient, electrical resistivity, thermal conductivity, and absolute temperature, respectively. Bi_2_Te_3_-based compounds have been widely commercialized as the best-performing room-temperature TE material.

The SGB were uniformly mixed in Bi_2_Te_3_ powder (0.5:99.5), and its effect on the TE properties of bulk nanocomposites was evaluated. Carbon burial sintering was used to prepare the bulk nanocomposites samples, as previously reported by [49]. Under the pressure of 40 MPa, the sample was cold-pressed in a stainless-steel die with a diameter of 12 mm and a thickness of 2–3 mm. Then, the sample was put in the alumina crucible surrounded by carbon powders and sintered at 474 K for 6 h.

The temperature dependence of the electrical resistivity for the samples is shown in Figure 7a. The electrical resistivity of pure Bi_2_Te_3_ (indicated by the black line) increases as the temperature increases from 320 K to 480 K, which indicates that the bulk materials exhibit a temperature dependence of the electrical resistivity described by metal-like or degenerated semiconductor behavior [50]. A significant drop in the electrical resistivity can be observed in the nanocomposite compared to the pure Bi_2_Te_3_ sample. This observation meets the statement made by Kaleem et al., who reported that graphene in SGB may account for the decrease in electrical resistance [51]. Figure 7b illustrates the Seebeck coefficients, S, for the samples. A significant increment in S is observed in the nanocomposite compared to the pure Bi_2_Te_3_ sample. It is more likely that the increase in the Seebeck coefficients, S, is due to the addition of SiO_2_ in SGB, which provides new barriers for charge carriers to overcome [52]. These early results suggest that this material has potential in optimizing TE properties for TE energy harvesters.

## 4. Conclusions

We successfully demonstrated an easy and fast method to directly grow graphene on SNP, called SGB, through a copper-vapor-assisted APCVD process. The detailed formation mechanisms for the SGB were proposed and investigated. Raman measurements support the existence of the sp^2^ carbon network. The morphology of the SGB is observed by FESEM as well as TEM. Although copper vapor is involved in the formation of SGB, the resulting SGB does not contain copper metal, as confirmed by FESEM-EDX. The investigation of the TE properties of the SGB as nanofillers in Bi_2_Te_3_ nanocomposites reveals significant improvements in the electrical resistivity and Seebeck coefficient as the temperature rises from 320 K to 480 K. We anticipate that our findings will spur additional research into using nanocomposites as a cost-effective strategy for significantly improving TE material performance at low operating temperatures.

## Figures and Tables

**Figure 1 nanomaterials-14-00618-f001:**
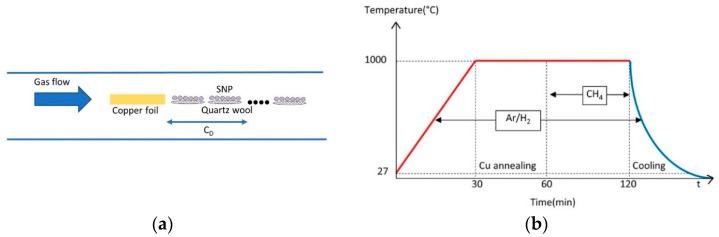
(**a**) Distance-dependent growth of SGB. Arrow shows the distance between copper foil and SNP. (**b**) Experimental protocol of SGB synthesis using APCVD. Arrow shows the duration of the gasses exposed during graphene growth.

**Figure 2 nanomaterials-14-00618-f002:**
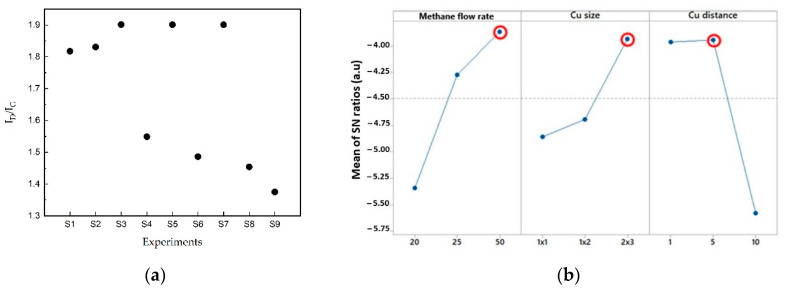
(**a**) I_D_/I_G_ derived from Raman spectra. (**b**) Main effects plot for SN ratios of SGB growth. The red circle shows the maximum value of Mean of SN ratios, therefore the value is the optimum parameter for SGB growth process.

**Figure 3 nanomaterials-14-00618-f003:**
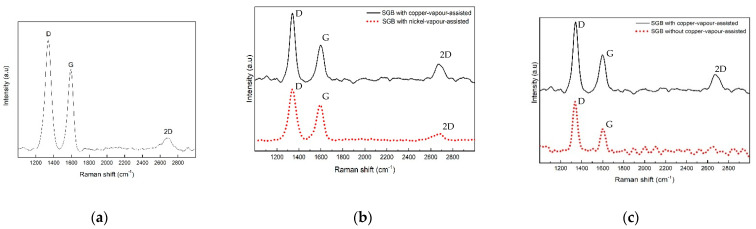
Raman spectra: (**a**) SGB at optimum condition; (**b**) comparison between SGB growth with copper and nickel as catalysts; (**c**) comparison between SGB growth with and without copper as catalysts.

**Figure 4 nanomaterials-14-00618-f004:**
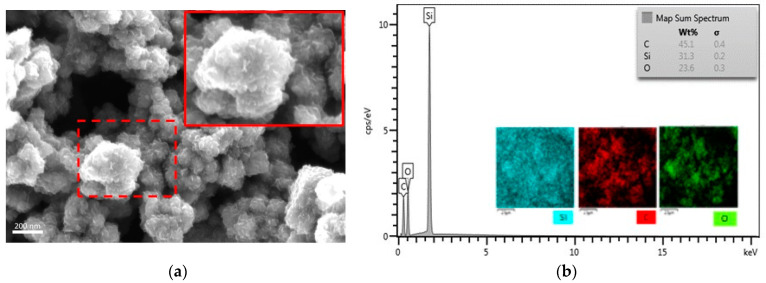
SGB at optimum condition: (**a**) FESEM image, the red lined shows zoomed specified spot shown by the small box with red dash line; (**b**) EDX spectra.

**Figure 5 nanomaterials-14-00618-f005:**
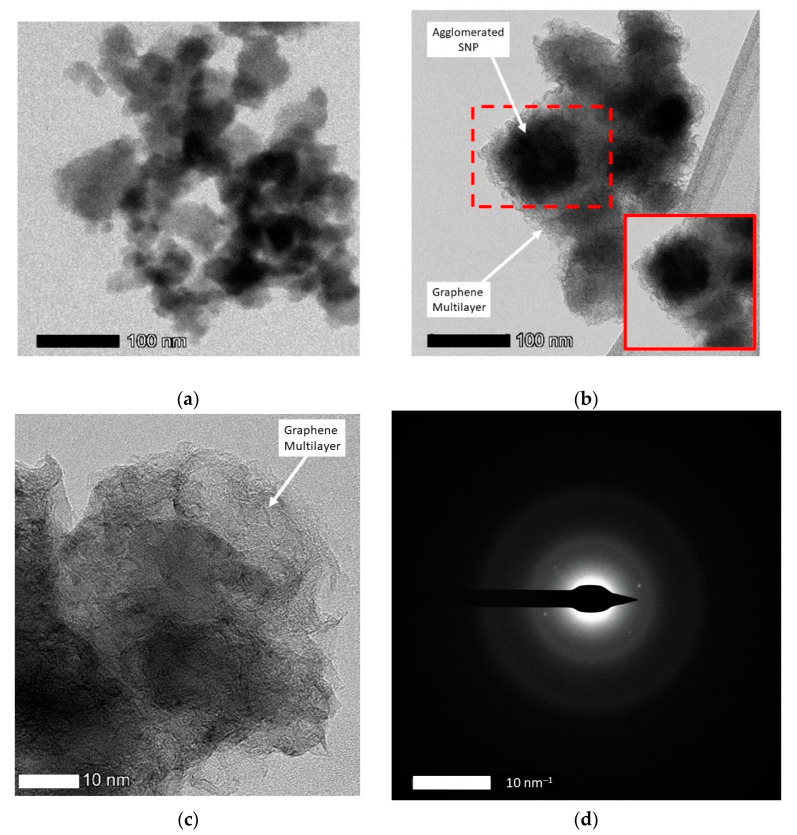
TEM micrographs of SNP and SGB at the optimum condition: (**a**) initial SNP; (**b**) SNP after coated with graphene; (**c**) SGB 3D sheet morphology; (**d**) the selected area of electron diffraction (SAED) pattern, obtained at 120 kV.

**Figure 6 nanomaterials-14-00618-f006:**
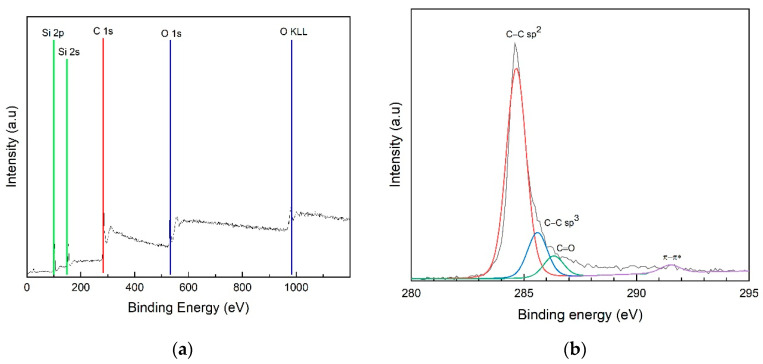
X-ray photoelectron spectroscopy (XPS) spectrum of SGB at the optimum condition: (**a**) the survey spectrum; (**b**) curve fit of C1s peak.

**Figure 7 nanomaterials-14-00618-f007:**
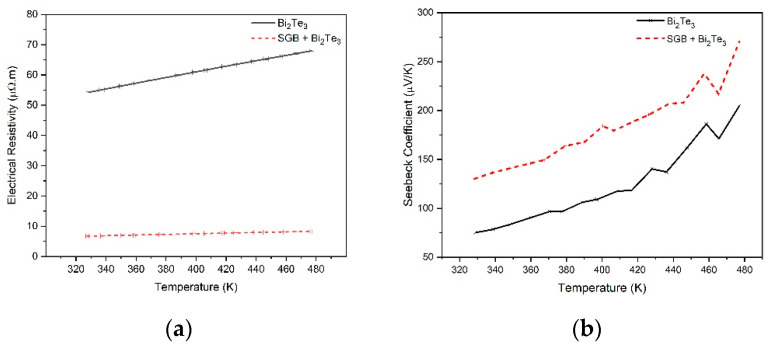
Temperature dependence of (**a**) electrical resistivity, *ρ*, and (**b**) Seebeck coefficient, S, of SGB/Bi_2_Te_3_ nanocomposites and pure Bi_2_Te_3_.

**Table 2 nanomaterials-14-00618-t002:** Taguchi-designed trial experiment for SGB APCVD.

Experiments	Factors		
	Methane Flow Rate, Φ_CH4_ (sccm)	Copper Foil Size, C_S_ (cm)	Copper Distance, C_D_ (cm)
S1	20	1 × 1	1
S2	20	1 × 2	5
S3	20	2 × 3	10
S4	25	1 × 1	5
S5	25	1 × 2	10
S6	25	2 × 3	1
S7	50	1 × 1	10
S8	50	1 × 2	1
S9	50	2 × 3	5

## Data Availability

Data are contained within the article.

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
