# Peer review of "Synthesis and Characterization of SiO2-Based Graphene Nanoballs Using Copper-Vapor-Assisted APCVD for Thermoelectric Application"

_nanomaterials, 2024, doi:10.3390/nano14070618_

Round 1

Reviewer 1 Report

In this paper, Graphene nanospheres were synthesized under atmospheric pressure assisted by copper vapor, and the experimental conditions were optimized by Taguchi method. Then, the performance of TE material was improved by filling it with TE material, with a simple and clear structure and certain innovation. This article is acceptable, but there are the following issues that need to be addressed.

1.       For line 182 and Figure 2a, based on the Raman spectrum in Figure 3a, with ID/IG>0, substituting the signal-to-noise ratio formula, the result should be positive. Why the ordinate of Figure 2a is negative?

2.       For the improvement of TE performance by SGB, there is no blank Scientific control to indicate that the source of contribution is Graphene powder or quartz particles or both of them.

3.       More description about the Taguchi-designed trial experiment should be given out, for example, why Methane flow rate of 20, 25 and 50 sccm was selected rather than uniform increase step.

4.       For the quality evaluation, ID/IG is one of the important factor but not the only one. More characterization are suggested to make the judgement standard reasonable. In addition, multi points/times results with error bar should be added to make the results reliable.

5.       Figure clarity should be strengthened, especially figure 6. Table 2 should be replotted and table 3 can be deleted since the information can be expressed in one sentence. 

Minor editing of English language required

Author Response

Dear Editors and Reviewer#1:                 

Thank you for your letter and the reviewer’s comments concerning our manuscript entitled “Synthesis and Characterisation of SiO2-based Graphene Nanoballs using Copper-Vapour-Assisted APCVD For Thermoelectric Application”. Those comments are all valuable and very helpful for revising and improving our paper, as well as the important guiding significance to our research. We have studied the comments carefully and have made the correction, which we hope meet with approval. All revised portions have been marked in yellow in the revised manuscript. The main corrections in the paper and the responses to the reviewer’s comments are as follows:

Reviewer 2 Report

The author describes the “Synthesis and characterization of SiO2-based Graphene Nanoballs using Copper-Vapour-Assisted APCVD For Thermoelectric Application”. This original article is quite interesting from a technological point of view. The author should revise their manuscript based on the comments and suggestions.  I recommended a Minor revision of the manuscript.

The Minor suggestion below:

  1. What is the main advantage of the APCVD method used for the fabrication of SiO2-based graphene nanoballs?
  2. The introduction is too wide, the author should focus on the originality of the current work.
  3. Why author choose the flow ratio of 78:2:50 (Ar/H2/CH4), any specific reason?
  4. The author should provide the low and high-magnification SEM images of SGB.
  5. The author should provide the SAED pattern of SGB.
  6. The figure quality is too poor and the author should improve the quality of the figure.
  7. The author should improve the grammatical and typo errors in the paper.
  8. The author should reformat the conclusion and provide a narrow conclusion for the paper.

Minor editing of the English language required

Author Response

Dear Editors and Reviewer#2:                 

Thank you for your letter and the reviewer’s comments concerning our manuscript entitled “Synthesis and Characterisation of SiO2-based Graphene Nanoballs using Copper-Vapour-Assisted APCVD For Thermoelectric Application”. Those comments are all valuable and very helpful for revising and improving our paper, as well as the important guiding significance to our research. We have studied the comments carefully and have made the correction, which we hope meet with approval. All revised portions have been marked in yellow in the revised manuscript. The main corrections in the paper and the responses to the reviewer’s comments are as follows:

Reviewer 3 Report

In this paper, the authors reported a method of synthesizing SiO2 based graphene nanospheres (SGB) by atmospheric pressure chemical vapor deposition (APCVD) assisted by copper vapor. This method should solve the contamination, damage, and high costs associated with silica-based indirect graphene synthesis.

This approach would offer a simple strategy to improve the TE performance of commercially available TE materials, which is critical for large-scale industrial applications. I believe that publication of the manuscript may be considered only after the following issues have been resolved.

1.       This work mentioned that the growth of Graphene does not require copper as a catalyst. What is the catalyst for this work? Suggest the author to explore more physical mechanisms.

2.       The author mentioned 9 schemes in the experiment of growing Graphene, but in the subsequent sample characterization, there is a lack of relevant characterization, so the author is suggested to supplement them.

3.       In order to better compare the relevant performance, it is necessary to supplement the relevant performance of SGB in Figure 6.

4.       The introduction can be improved. The articles related to some applications of graphene and graphene oxide materials should be added such as Results in Physics 48, 2023, 106420; Micromachines 2023, 14, 953; Commun. Theor. Phys. 2023, 75, 045503; Optics Express, 30(20), 35554-35566, 2022.

5.       Please check the grammar and spelling mistakes of the whole manuscript.

Minor editing of English language required

Author Response

Dear Editors and Reviewer#3:                 

Thank you for your letter and the reviewer’s comments concerning our manuscript entitled “Synthesis and Characterisation of SiO2-based Graphene Nanoballs using Copper-Vapour-Assisted APCVD For Thermoelectric Application”. Those comments are all valuable and very helpful for revising and improving our paper, as well as the important guiding significance to our research. We have studied the comments carefully and have made the correction, which we hope meet with approval. All revised portions have been marked in yellow in the revised manuscript. The main corrections in the paper and the responses to the reviewer’s comments are as follows:

Round 2

Reviewer 3 Report

accept

Author Response

We have made corrections as requested by the reviewer. 

Please find our response in our attached revised manuscript.
